# Microstructure and Soil Wear Resistance of a Grey Cast Iron Alloy Reinforced with Ni and Cr Laser Coatings

**DOI:** 10.3390/ma15093153

**Published:** 2022-04-27

**Authors:** Marta Paczkowska, Jaroslaw Selech

**Affiliations:** Department of Transport and Civil Engineering, Institute of Machines and Motor Vehicles, Poznan University of Technology, Piotrowo 3, 60-965 Poznan, Poland; marta.paczkowska@put.poznan.pl

**Keywords:** laser modification, nickel, chromium, grey iron, tribological behaviour properties, agriculture machine parts

## Abstract

The goal of the presented investigation was to assess the impact of surface laser modification with the implementation of nickel and chromium on the microstructure and tribological behaviour of grey iron. Surface laser modification consisted of remelting the surface layer with simultaneous implementation of selected elements. In the first variant of treatment only nickel was implemented and in the second one, a combination of nickel with chromium together. This treatment was performed on an agriculture machine part made of grey iron and working in intensive friction conditions. The constituted surface layer was characterized by about 0.45 mm of depth and a 160 mm^2^ area of the most exposed to wear of the treated part. In the case of both types of variants, the achieved surface layer microstructure was identified as homogenized with small grains. It involved nickel in the first variant of modification and nickel and chromium in the second one. The attained microstructure with nickel addition was characterized by nearly 800 HV0.1 of hardness (a 3.6-fold increase in comparison to its core material). The approximate hardness of 900 HV0.1 was achieved in the case of the microstructure enriched with nickel and chromium (over a 4-fold increase in comparison to the core material). The roughness of the surface after laser modification was reduced (nearly 3-fold) in comparison to the original surface of the part that was characterized by quite substantial coarseness. The wear test showed that Ni and Cr laser coatings increased resistance to abrasive wear resulting from the modification of the microstructure by the formation of martensite and grain fragmentation. Laser modified parts had a 2.5-fold smaller mass loss than untreated parts. Both types of performed variants: with the implementation of nickel and a combination of nickel and chromium gave comparable effects.

## 1. Introduction

An increase in surface durability of many machine parts, including agriculture parts operating in the soil can be achieved by modification of their surface layer characteristics such as wear and corrosion resistance. It needs to be taken into account that recent working speeds of such machine parts have at least tripled [1]. Thus, problems occur with accelerated wear resulting in downtime and loss of time due to the need for replacement with a new part. Any downtime can mean high economic costs and lower yields. The recipient of the tilling set is focused on its reliable and, preferably, maintenance-free use during the agrotechnical period. The main challenge for fast-wearing elements is to find a treatment that would modify the surface layer and significantly extend the life of machine elements operating in very difficult conditions.

The growing need for agricultural machines requires the development of new technological solutions to address problems of wear of machine parts, designed for work in a soil medium [2,3,4,5,6,7,8]. To extend the intervals between worn machine part replacement and generally to raise their durability, appropriate surface treatments should be applied [9,10,11,12,13]. Laser surface modification could be one such treatment, because, by using this method, it is possible to effectively improve the microstructure and properties of the surface layer of parts made of metal alloys along with their cast alloys, like grey irons [14,15]. There are many methods of laser surface modification. Laser processing allows the surface layer to be melted down with the simultaneous addition of alloying elements to it. The resulting microstructure is usually highly fragmented and often contains martensite. In some cases, it is similar to ledeburite, for example, containing primary cementite spines. In the literature, there are many examples of grey cast iron surface modifications made with the use of a laser beam, which significantly changes the selected parameters of the surface layer. The laser implementation of selected elements, i.e., carbon, boron, tungsten or chromium on the surface of nodular iron, resulted in the formation of a microstructure containing a lot of carbides with a very high hardness reaching 1200 HV0.05 [16]. The increase in hardness and increased corrosion resistance were also found after the implantation of copper in cast iron [17]. This was observed in the case of carbon implementation into the grey iron [18]. In turn, a 3.5-fold decrease in the wear rate of parts made of nodular iron (when compared to the only hardened parts) was proved after laser implementation of boron [19]. The research [20] analysing the influence of this kind of modification, showed that it is possible to obtain a small-grained microstructure with martensite and iron borides that can increase the hardness of the modified layer by six times. To increase the hardness and corrosion resistance, a commonly used element is chromium. It could also be implemented during laser treatment. For example, a 4-fold increase in the hardness of the surface layer was noted in the research after such treatment. [5,21,22,23]. Additionally, some heat resistance can also be achieved by chromium implementation [24]. However, it needs to be noted that the positive effect of the laser implementation of chromium does not always appear. In the research [21] it was found that such a surface layer reduced the wear resistance of treated parts. Another interesting element for laser implementation is nickel, which is generally known for increasing ductility, toughness and corrosion resistance of ferrous alloys. In the case of laser modification of grey iron using this element together with chromium, an increase in corrosion as well as wear resistance has been noticed [22,25]. In other research [26,27,28] it was observed that the microstructure of the surface layer modified in such a way blocked thermal crack propagation. Based on the scientific papers cited above, it could be indicated that nickel implementation to the surface layer of grey cast iron machine parts operating in soil, using laser heating, should help in the reduction of wear. Despite some noted problems with layers containing chromium, that could appear, the effect of this element (especially together with nickel) also seems to be worthy of analysis [29,30,31]. The goal of the presented investigation was to assess the impact of the surface laser modification with the implementation of nickel and chromium on the microstructure and tribological behaviour of grey iron. There is no data regarding the influence of nickel or nickel and chromium implementation during laser treatment onto the surface layer of a part of a cultivator coulter on the resistance to abrasive wear.

## 2. Material and Methods

The test object was the cultivator coulter flap of a mechanical drill seeder (Figure 1). This kind of cultivator coulter flap was designed to be an easy-to-remove and replace part. Such cultivator coulter flaps are characterized by average dimensions of 60 × 235 × 18 mm and their approximate average mass is about 1175 ± 75 g. These flaps are flat-shaped. The material used for coulter flap production is grey cast iron with flake graphite and mainly perlite in the matrix (Figure 2). There is no through-heat treatment of coulters except surface treatment consisting of a fast cooling process of their surface layer to create a transformed ledeburite microstructure. The strength properties of this material are determined by the producer according to Standards. Surface laser modification consisted of remelting the surface layer with the simultaneous implementation of elements. For the first variant of modification, nickel was used as an alloying element and in the other one, the combination of nickel with chromium (in the composition of 1:1 in weight) was applied. The particle size and the purity were <45 µm and ≥99.8% for nickel and <1 μm and ≥99.0% for chromium, respectively. The coating was mixed with one or more elements and water glass as a bounding substance, and distilled water. The heating using a dual diode TRUDISK 1000 laser device was performed in such a way as to remelt the surface layer of a piece of the cultivator coulter tip. It was taken into account that all coulter flaps work at a depth of about 5 cm ± 2 cm. The laser beam power fluence was 54 J/mm^2^ and its collimation was equal to 12 mm. Such a value of this parameter caused the diameter of the laser track of 1.16 mm. The distance from the centers of particular laser tracks was 1.1 mm. The laser beam scanned the surface to create an area of modified surface layer of approximately 160 mm^2^ near the edge of the coulter. To carry out the research of the microstructure of the modified surface layer Vega 5135 and MIRA3 Tescan scanning electron microscopes (Tescan, Brno, Czech Republic) were used. A metallographic study was performed on samples etched with nitric acid and Kalling’s reagent. Elements were evaluated by the PGT Prism 200 Avalon EDS microanalyzer (Princeton Gamma-Tech, Princeton, NJ, USA). The surface profile estimation was performed with a Zeiss contact profilometer. Hardness measurements were performed with a Zwick 3212 hardness device using the Vickers method, with a load of 100 g according to the standard PN-EN ISO 6507-1:1999. The evaluation of wear resistance in a sandy medium was carried out using a special tribological tester (Figure 3). Conditions in this tester imitate operation conditions (kind of movement and kind of friction) of cultivator coulters during seeding. Such a device was also used in the research presented in [32,33]. Using a holding-arm, coulters are placed in the bowl (with a diameter of 1.6 m) filled with sandy medium (Figure 4). In this experiment, 5 cultivator coulters were modified with nickel and 5 with the combination of nickel and chromium. Five coulters also untreated by laser modification were tested. The travelling distance for each cultivator coulter was calculated in such a way as to reflect the seeding of seeds on a field of 35 hectares. Samples from the wear test are presented in Figure 4. An abrasive material, silica sand with a grain fraction of 0.2–0.3 mm, and hardness at 995 ± 10% HV was used. The sand was selected in such a way that the shape of grains and the degree of reeling matched the similarity to soil sands, in accordance with the PN-EN 933-1:2001 standard. In order to achieve the right fraction and dispose of dust and organic pollutants, the sand was flushed, and a sieve analysis was performed, according to the PN-EN 933-4:2008 standard. An examination of the grain size indicated that the share of pollution is <3%. The sand moisture content was about 10% dry weight, and it was determined by measuring the weight of solid-phase dried at 105 °C. The wear experiment results were determined (except for the surface profile analysis mentioned above) by measuring the masses of coulter flaps before and after the test using a precision mass scale. Because there was a difference in linear velocities at various distances from the rotation axis of the bowl, the arm holding the test samples was inverted halfway through the test, and the rotation direction was changed as well. The result was that the wear of each of the three samples tested simultaneously was more uniform. In order to reflect actual working conditions as closely as possible, the samples were placed at intervals of approx. 12 cm from one another to ensure that the abrasive medium flowing over the sample is not affected by the adjacent sample. The measured linear velocities calculated for the three samples at different distances from the rotation axis of the bowl were about as follows: 6.4 km/h, 5.4 km/h, and 4.3 km/h. Thanks to the inversion of the arm holding the samples, it was possible to ensure that the distance travelled by all coulters was identical, irrespective of their position on the holding arm [15]. Abrasive wear was the main mechanism for material wear in soil. The soil mainly contained a sand fraction from 0.2 to 0.3 mm—about 90%, and a smaller fraction from 0.01 to 0.2—about 8%, and the rest was a fraction of dust. The increased portion of coarse grains was responsible for microcutting and grooving, and the portion of fine fractions for microcutting.

## 3. Results and Discussion

In the case of all cultivator coulters modified surface layer-specific for a laser treatment was detected. This specific modification appears as a melted zone. This zone was created on the part of the ‘edge’ of the cultivator coulter. The modified surface on the coulter is visible in Figure 5 for laser implementation of nickel (a) and composition of nickel and chromium (b). The size of the modified surface on the edge was approx. 160 mm^2^.

Examples of the surface profiles of a modified part are presented in Figure 5. As could be noticed, such treatment can change the stereometric structure of the surface quality of the cultivator coulter. The roughness (taking into account the arithmetic average roughness Ra) of the surface after both variants of laser modification was characterized by a nearly 3-fold decrease in comparison to the original surface of the cultivator coulter (Figure 6). However, there was some higher scatter in results in the case of coulters after the variant of modification consisting of implementing chromium and nickel together. The mean roughness depth (Rz) for the untreated surface was 55 μm, after nickel implementation it was 17.5 μm and after implementation of a combination of nickel and chromium was 21 μm. In both cases, the roughness depth after laser treatment was smaller than the roughness depth of an untreated surface. The surface roughness is usually improved by laser alloying in the case of treating a rough surface such as those of cultivator coulters [34]. Nevertheless, in more cases when the roughness of the base surface is small, laser remelting causes an increase of the roughness parameters—for example in the research regarding the laser nitriding of the NiTi alloy [35].

A microscopy analysis of a cross section of the surface layer showed that after laser modification an alloyed zone appeared. It was observed in the case of the nickel implantation variant as well as nickel and chromium. Such a zone is characterized by a very fine-grained microstructure that can be seen from the surface to a depth of approx. 450 μm in Figure 7a,b. This is due to a large number of crystallisation nuclei that appear during rapid solidification from the liquid state. This process also takes place during laser remelting without alloying. Such a microstructure is also much more homogeneous than the microstructure of transformed ledeburite microstructure (visible in Figure 8) or core material microstructure consisting of flake graphite in the pearlite matrix.

More detailed SEM observation of the alloyed zone microstructure in 2k× magnification revealed a morphology characteristic for this kind of laser modification which is based on the presence of a dendritic microstructure in the whole modified zone. This is due to the melting and solidification processes of the surface layer. Such a process takes place as well in the case of only laser remelting. The difference between only remelting and alloying is the possibility of forming some new phases containing implementing elements in case of alloying. The dendritic microstructure is visible in both cases of the performed laser treatment variants (Figure 9a and Figure 10a). A microstructure study allowed us to conclude, that during the second part of the modification solidification of the surface layer, austenite dendrites are crystallizing directly from the liquid state of a new mixed alloy (consisting of base material and alloying additions). As a consequence of the rapid cooling, hardening takes place (it also takes place in the case of laser remelting only). Non-equilibrium solidification of new alloy takes place. A similar microstructure was achieved after laser cladding using nickel on ductile cast iron [2] improved resistance to corrosion. But resistance to wear has not been studied. A higher magnification of the modified layer (10k×) achieved by SEM analysis (Figure 9b and Figure 10b) allowed us to detect the results of rapid solidification during the laser treatment. Laser hardening caused the formation of the martensite phase. Martensite appears in the case of laser remelting and in most cases of laser alloying –except for implementing of cobalt, for example [36]. Martensite was especially clearly observed as needles in the dendritic microstructure (observed in figures with lower magnification—Figure 9b and Figure 10b). Between the martensite needles retained austenite is present. Such a microstructure has also been observed, for example, in research referring to the laser implementation of silicon [37,38].

The SEM observation of the surface layer of the coulter with no laser treatment in higher magnification (2k×) that can be seen in Figure 11 revealed a transformed ledeburite microstructure characteristic of white cast iron. Such a microstructure consists of pearlite (visible as a cementite and ferrite plates) and eutectic. It is worth emphasising that in comparison to the surface layer after laser treatment, this microstructure is much more coarse-grained and heterogeneous. Additionally, grains are characterized by sharp ends which is not conducive to wear resistance.

An X-ray microanalysis of the laser modified layers showed that the investigated microstructures are enriched with nickel (in case of laser implementation of nickel) and with nickel and chromium (in case of laser implementation of a combination of those elements). An EDS spectrum was made on the section from the surface to the core material in both analysed variants. The distribution in the section of basic elements is shown in Figure 12. Except for elements that are present in grey irons naturally like iron, carbon or silicon, nickel was additionally detected in the case of both variants of laser modification (Figure 9a,b) and chromium was detected in the surface layer after laser implementation of a combination of nickel and chromium. The EDS analysis confirmed the depth of the alloyed layers observed during the microstructure analysis. In both cases, it was about 0.45 mm. The average amount of alloying elements was about 3 wt.% of Ni after implementation of this element, and about 2 wt.% of Ni and 2.5 wt.% of Cr in the case of implantation of nickel and chromium. The examples of the EDS spectrum are presented in Figure 12. The chemical composition is presented in Table 1.

The hardened microstructure of the layer enriched with implemented elements was characterized by an increase in hardness in comparison to an untreated coulter with a white cast iron microstructure in the surface layer (Figure 13). The addition of nickel allowed a hardness of nearly 800 HV0.1 to be achieved and in the case of the addition of nickel and chromium, the mean hardness of the surface layer was approx. 900 HV0.1. So, 3.6- and the nearly 4-fold increase was noticed. For example, after the implementation of Cu-Ti-Ni by laser cladding on grey cast iron was improved 2–3 times in comparison to the material of the substrate [39].

The performed wear tests proved that laser modification of the surface layer of the coulters radically decreased its mass loss (Figure 14). It needs to be stated that the area of modification was relatively small. Probably it was enough to reduce the wear. A 2.5-fold decrease in mass loss was noticed in the case of two performed variants. In the case of the investigated type of friction, the addition of chromium to the surface layer did not change the wear behaviour in comparison to the modification consisting of implantation to the surface layer of nickel only.

Therefore, by changing the microstructure of the surface layer it is possible to improve the wear resistance of machine parts working in soil mediums such as cultivator coulters. By achieving hardness of approx. 900 HV0.1 after laser implementation of nickel and chromium and nearly 800 HV0.1 after implementation of nickel only, a 60% mass loss decrease could be achieved by modification of the surface of only a small area and on only one side of the edge of the coulter.

It needs to be noticed that the not very large increase of the surface layer hardness in comparison to the coulters only with a white cast iron microstructure in the surface layer caused a significant increase of their wear resistance. The mass loss reduction of the treated parts working in fruition conditions is a consequence not only of increased surface hardness but also a result of a significantly modified, more uniform microstructure with small grains with less sharp ends that are stress concentrators. Hence, the quite high hardness of the surface layer of the cultivator coulters made of transomed ledeburite (not laser treated) does not turn into proportionally greater wear resistance.

A macroscopic examination revealed that the tribological test caused some changes in the appearance of the surface in all cases. Mostly coulters without performed laser modification seemed to be smoother (Figure 15) in comparison to their state before the test (Figure 5). The edge of the coulter became rounded.

The analysis of the surface profiles of the investigated coulters after the tribological test showed that the arithmetic average roughness, as well as mean roughness depth, decreased for coulters without laser modification, Ra was about 7.5 μm, and Rz 36 μm. The changes in surface roughness parameters for coulters with laser modification were slight after the tribological test. The arithmetic average roughness was 3.5 μm in the case of surface implemented with nickel and over 4 μm in the case of surface layer implemented with a combination of nickel and chromium and the mean roughness depth was 19 μm and 21 μm, respectively. The surface laser treatment increased the surface roughness of grey cast iron due to the laser ablation by the instant high temperature of a laser beam. The microhardness and elemental composition along the cross-section in both cases (Ni and Ni + Cr) were also analysed, which shows that the hardness was increased after laser modification treatment. The friction coefficient and wear resistance of grey cast iron was also improved after laser modification.

## 4. Conclusions

The performed research allows us to state, that it is possible to effectively modify the surface layer of grey iron machine parts, like agriculture parts operating in the soil to increase their resistance to wear by laser surface modification. It was achieved by implementing elements such as nickel and chromium into the surface layer that was originally made of white cast iron.

On the basis of the conducted research, it can be concluded that:a size of the formed surface layer was characterized by about 0.45 mm of depth and 160 mm^2^ of area (on the edge of the cultivator coulter);a modified area after laser alloying shown after nickel as well as nickel and chromium implementation was characterized by hardened microstructure (with martensite), higher homogeneity and fine grains;nickel, as well as nickel and chromium presence in laser modified surface layer, has been proved;the hardness of the layer only with nickel addition was nearly 800 HV0.1 (so it was a 3.6-fold increase in comparison to its core material);the hardness of the layer with the addition of a composition of two elements: nickel and chromium was approximately 900 HV0.1 (an over 4-fold increase);the surface geometrical structure was changed after laser treatment (Ra parameter was nearly 3-fold reduced as a consequence of laser modification in the case of both performed variants with one and two elements);the laser modified parts were characterized by a 2.5-fold smaller mass loss than the untreated parts;the implementation of nickel into the surface layer as well as implementation together with chromium caused comparable results in the case of wear resistance to friction in the soil medium;the roughness after the tribological test was changed mainly for unmodified parts.

As results showed, there is no difference in wear resistance between the implementation of only nickel and nickel and chromium. Consequently, it could be stated that there is no need to add chromium into the surface layer of the coulter (except for the possibility of increasing corrosion resistance if it were proved in such a case).

It is worthy to underline that the area of the modified cultivator coulter was relatively very small but the effects of resistance to wear appeared significant. Thus, it could be stated that a slight change of the surface layer made by laser surface modification of machine parts operating in an abrasive medium like in the case of agriculture can change their life considerably.

It could be concluded that hardness above 800 HV0.1 seems to be enough to achieve quite good wear resistance results. It is more important to create a uniform and fine crystalline microstructure, as was achieved in the presented research. As it appeared, the laser alloying in the case of the analysed machine parts decreases the surface roughness due to the high roughness of the base surface. However, it is worth underlining that there is no need to achieve a smooth surface in the case of parts working in soil. Therefore, a superfinishing process in the case of a cultivator coulter is not required.

The parameters of the performed laser treatment such as the laser beam power fluence of 54 J/mm^2^ and its collimation of 12 mm were suitable to achieve the modified surface layer that increased swear resistance.

Furthermore, it can be attained by economically reasonable repair of worn machines with wide use in the regeneration processes. Consequently, it is possible to reduce material costs.

The studies carried out also have some limitations. They were carried out in constant laboratory conditions, i.e., constant soil moisture and pH, constant speed and working depth of the tested samples, and wall medium (in this case river sand) with a specific fraction. Field conditions vary considerably and depend on many factors, such as soil type, compactness, moisture, contamination, soil pH, etc. Under such conditions, the wear intensity of the test samples could differ from the obtained laboratory results and would depend mainly on the previously mentioned soil parameters. Therefore, the directions of further research should take into account all these factors, and perhaps even be carried out in the field.

Another important factor that may limit the use of the aforementioned laser treatments is the cost of applying the appropriate Cr and Ni coating in laboratory conditions, which is several times higher than the purchase of a new spare part made of grey cast iron. If the indicated improvements to the coulters are to be introduced on an industrial scale and reduce the costs of modification of top layers, this process should be automated directly at the manufacturing plant of the spare part. However, this could present some difficulties due to the lack of appropriate technical facilities and the high cost of purchasing a laser coating apparatus.

## Figures and Tables

**Figure 1 materials-15-03153-f001:**
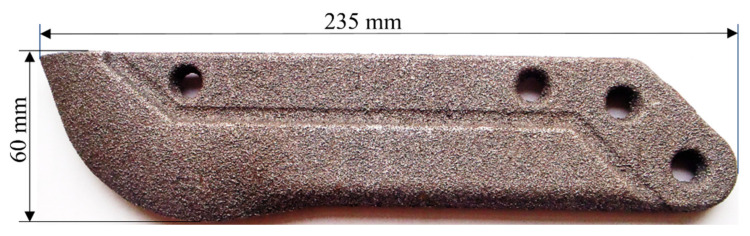
The cultivator coulter.

**Figure 2 materials-15-03153-f002:**
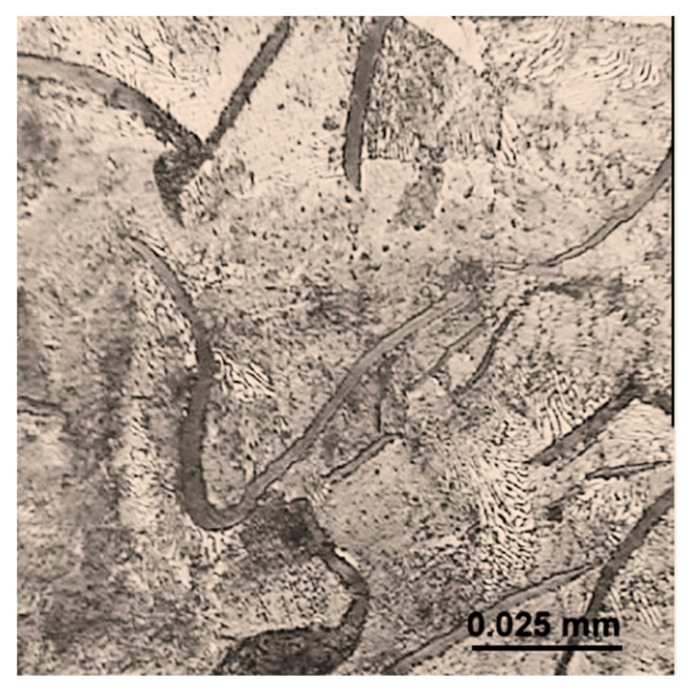
A microstructure in the core of the cultivator coulter.

**Figure 3 materials-15-03153-f003:**
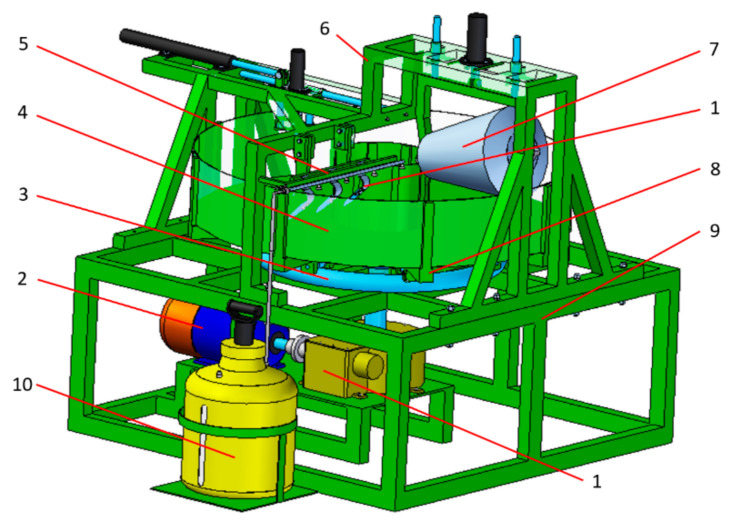
The wear test device consists of: transmission (1), motor (2), running rail (3), bowl (4), sample holder (5), supporting frame (6), compacting roller (7), bowl frame (8), main frame (9), water tank (10) [34].

**Figure 4 materials-15-03153-f004:**
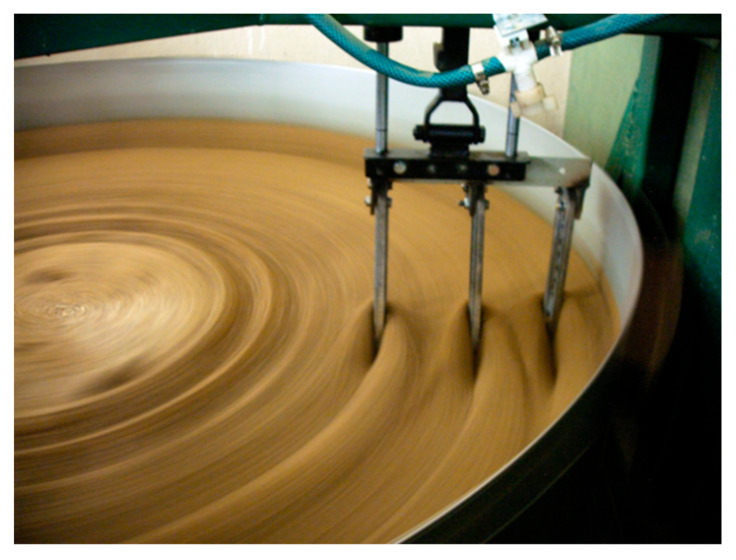
Samples during the tribological test.

**Figure 5 materials-15-03153-f005:**
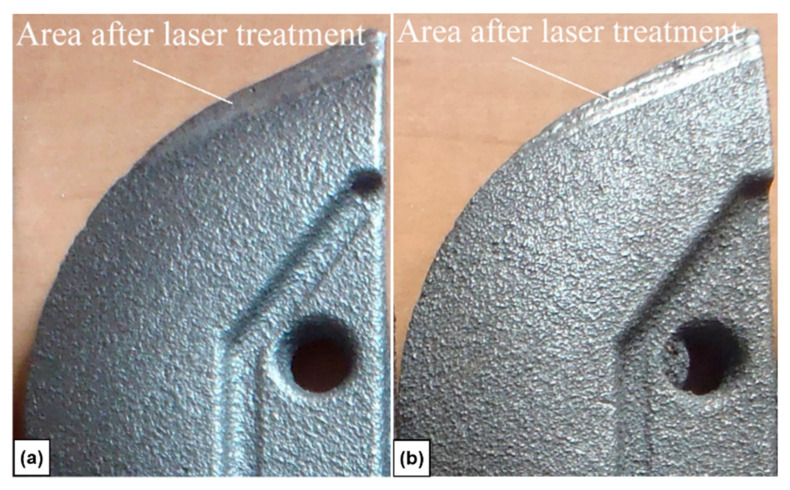
The cultivator coulter after laser implementation of nickel (**a**) and a combination of nickel and chromium (**b**).

**Figure 6 materials-15-03153-f006:**
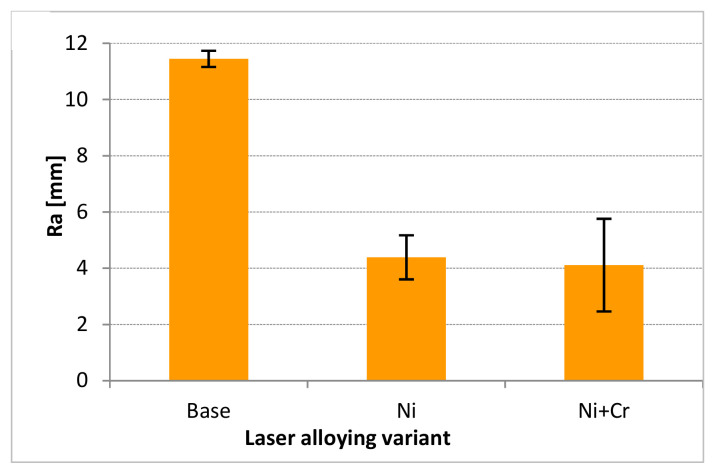
R_a_ parameter of the investigated cultivator coulter for base and after laser implementation of nickel and a combination of nickel and chromium.

**Figure 7 materials-15-03153-f007:**
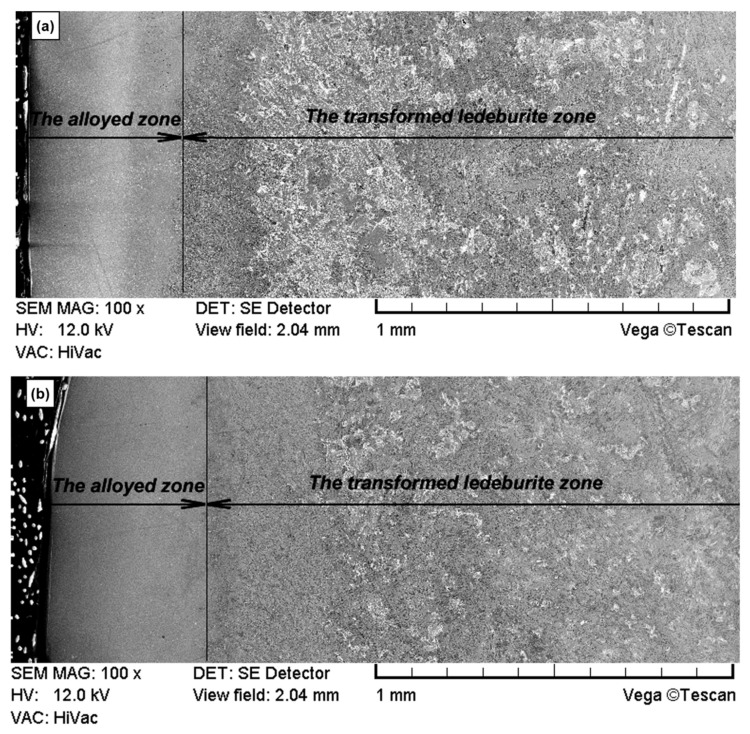
Microstructure of the surface layer after laser implementation of nickel (**a**) and a combination of nickel and chromium (**b**) (scanning electron microscope, etched with nitric acid).

**Figure 8 materials-15-03153-f008:**
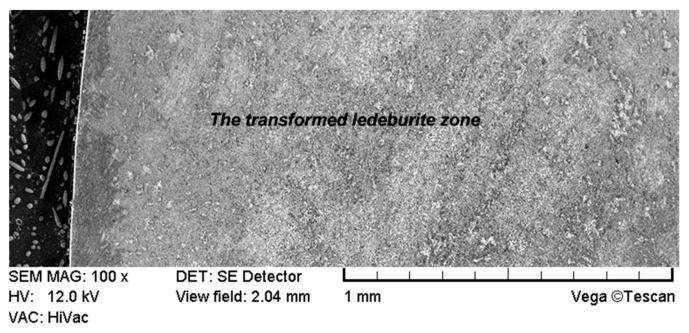
Microstructure of the surface layer of the untreated cultivator coulter (scanning electron microscope, etched with nitric acid).

**Figure 9 materials-15-03153-f009:**
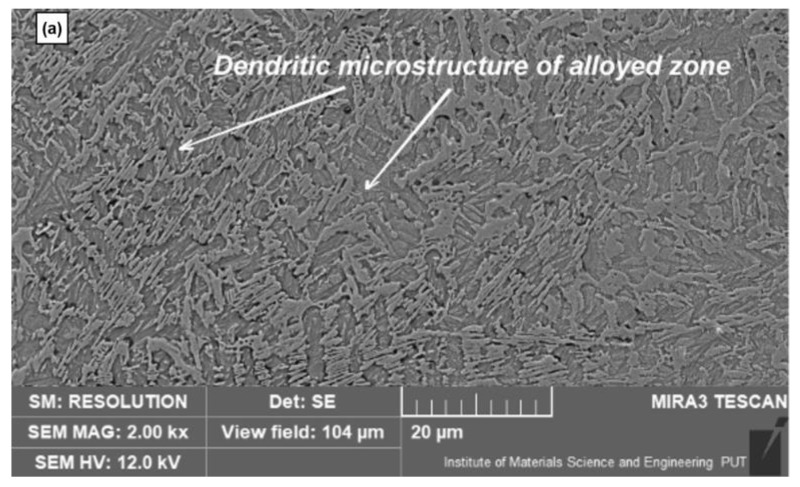
The microstructure of the surface layer after laser implementation of nickel; (**a**) SEM micrographs at 2k×; (**b**) SEM micrographs 10k× magnification.

**Figure 10 materials-15-03153-f010:**
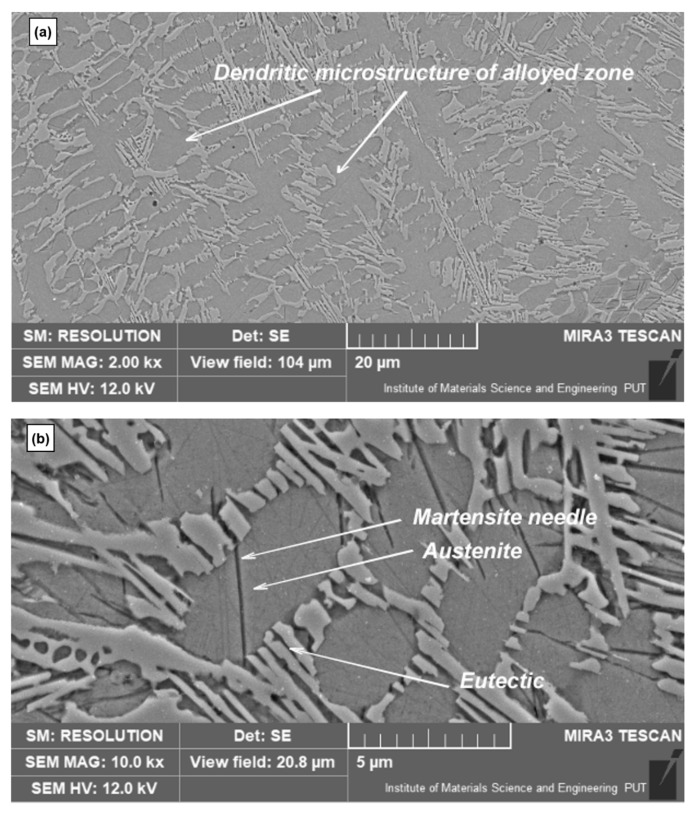
The microstructure of the surface layer after laser implementation of a combination of nickel and chromium; (**a**) SEM micrographs at 2k×; (**b**) SEM micrographs 10k× magnification.

**Figure 11 materials-15-03153-f011:**
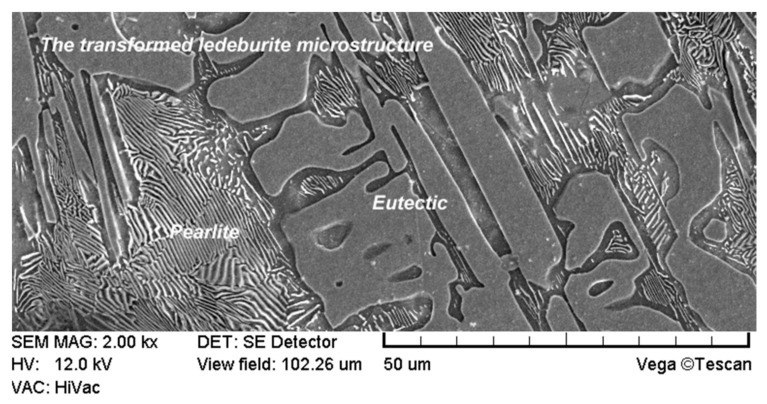
The microstructure of the surface layer of the cultivator coulter without laser modification.

**Figure 12 materials-15-03153-f012:**
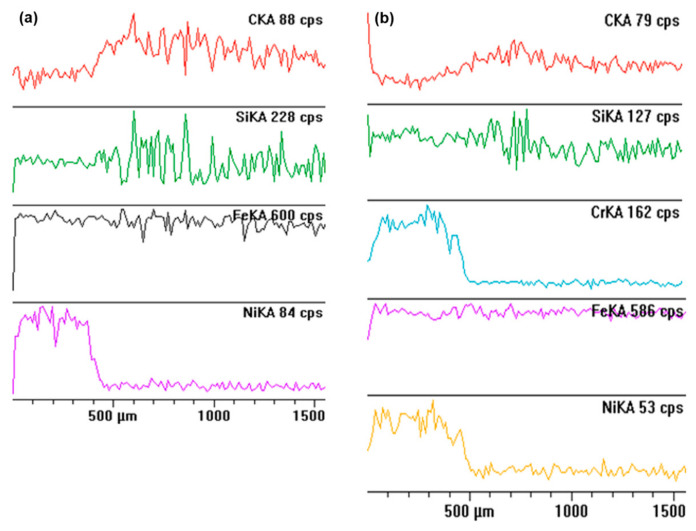
The distribution of investigated elements on the section from the surface to the core material for the cultivator coulter after laser implementation of alloyed nickel (**a**) and a combination of nickel and chromium (**b**) achieved with an EDS microanalyzer.

**Figure 13 materials-15-03153-f013:**
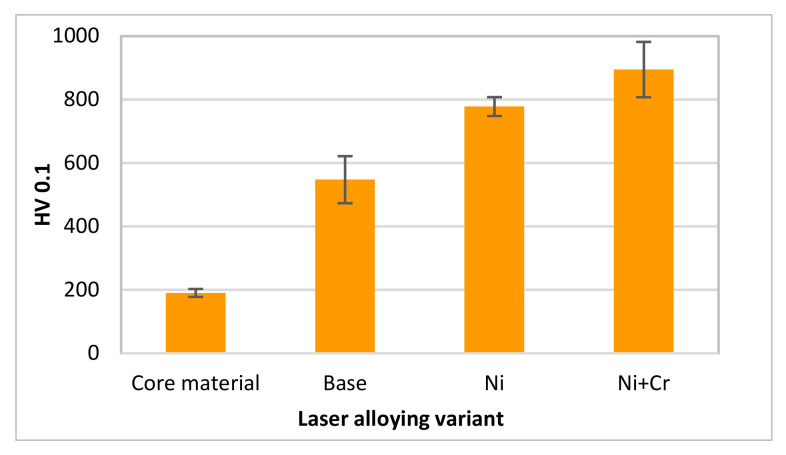
The average value of surface layers hardness.

**Figure 14 materials-15-03153-f014:**
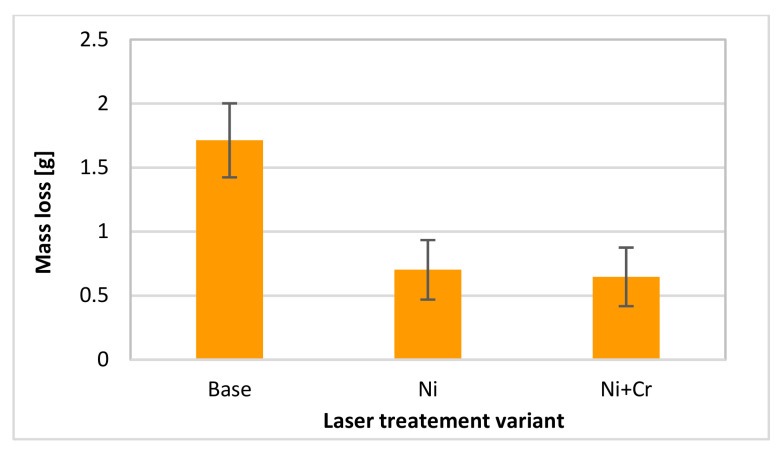
Mean weight loss after wear test.

**Figure 15 materials-15-03153-f015:**
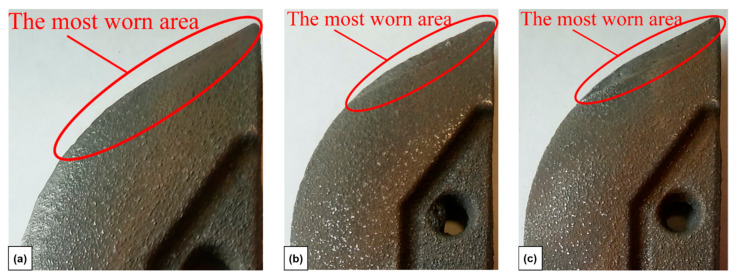
The cultivator coulter (after tribological test): without laser modification (**a**), after laser implantation of nickel (**b**) and combination of nickel and chromium (**c**).

**Table 1 materials-15-03153-t001:** The percentage of elements after implementation of nickel and combination of nickel and chromium.

Element	Ni Implementation		Ni + Cr Implementation	
	wt%	δ	wt%	δ
Fe	93.8	0.2	93.1	0.2
Cr	-	-	3.3	0.1
Ni	3.9	−0.2	1.9	0.2
Si	2.4	0	1.7	0

## Data Availability

The data presented in this study are available on request from the corresponding author.

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
