# Peer review of "Microstructure and Soil Wear Resistance of a Grey Cast Iron Alloy Reinforced with Ni and Cr Laser Coatings"

_materials, 2022, doi:10.3390/ma15093153_

Round 1
Reviewer 1 Report
The paper discusses the wear resistance improvement using two materials by laser implantation. The paper flow is good, but the results section needs more results to support the wear resistance improvements. Besides Figure 16, I did not see any significant change, or maybe this figure needs a closer look and more precise explanation.
Author Response
Dear Reviewer,
We would like to thank the Reviewer for valuable comments, which in our opinion contributed to increasing the quality of the paper. All suggestions and comments have been carefully thought out, and the corresponding revisions have been made to the paper. All changes in the article are marked blue red color in revision mode.
In addition, we have supplemented the incorporated changes with specific answers given below.
Yours sincerely
Authors
Comments and Suggestions for Authors
The paper discusses the wear resistance improvement using two materials by laser implantation. The paper flow is good, but the results section needs more results to support the wear resistance improvements. Besides Figure 16, I did not see any significant change, or maybe this figure needs a closer look and more precise explanation.
We added more results to support the wear resistance improvements. We also explained in detail Fig. 16. (after revision Fig. 15.)

Reviewer 2 Report
The manuscript is dealing an investigation on the impact of laser surface modification on the microstructure and wear behavior of grey iron. However, following shortcomings are presented in the manuscript and which has to be rectified before further processing. I request mandatory revision, as listed below, please do not simply respond but revise manuscript.
- Manuscript title is not informative. Authors should modify the title as an informative one with minimal wordings.
- Introduction, especially literature survey is not sufficient for problem definition. Include similar recent research works in this section.
- For readers to quickly catch the contribution in this work, it would be better to highlight major difficulties and challenges, and authors' original achievements to overcome them, in a clearer way in Introduction section.
- It is suggested to highlight the limitations of this study, suggested improvements of this work and future directions in the conclusion section.
- The properties of grey cast iron are well-established and no need to mention with a separate table. Therefore, Table 1 should be removed.
- Section 2 (Materials and methods) is described as point by point. It is not necessary to write the coating methods and their investigation procedure in point by point. It should be single paragraph (Descriptive).
- Figure 2 is not mentioned in the manuscript. Similar mistakes have to be rectified throughout the manuscript.
- The manuscript contains many SEM images, however, the detailed explanation about the inferences from the micrographs are missing. It should be discussed elaborately.
- The visual findings of the SEM micrographs should be mentioned in the images itself. Unfortunately, none of the SEM images contains the inference. It should be incorporated.
- The results and discussion are not clearly dealt the outcomes of the proposed work. The authors should explicitly state the novel contribution of this work, the similarities and the differences of this work with the previous publications in this section.
- Moreover, an extensive language editing is required to improve the quality of the manuscript. Authors are urged to rewrite the manuscript with a native English speaker.
Please note that the comments are intended merely to assist the authors in improving the manuscript and ensuring that published papers are of the highest quality. They are in NO WAY intended to discourage or demean the authors personally.
Author Response
Dear Reviewer,
We would like to thank the Reviewer for valuable comments, which in our opinion contributed to increasing the quality of the paper. All suggestions and comments have been carefully thought out, and the corresponding revisions have been made to the paper. All changes in the article are marked blue red color in revision mode.
In addition, we have supplemented the incorporated changes with specific answers given below.
Yours sincerely
Authors
The manuscript is dealing an investigation on the impact of laser surface modification on the microstructure and wear behaviour of grey iron. However, following shortcomings are presented in the manuscript and which has to be rectified before further processing. I request mandatory revision, as listed below, please do not simply respond but revise manuscript.
- Manuscript title is not informative. Authors should modify the title as an informative one with minimal wordings.
We corrected the title of the article to be more informative
Microstructure and wear resistance of a grey cast iron alloy reinforced with Ni and Cr laser coatings
- Introduction, especially literature survey is not sufficient for problem definition. Include similar recent research works in this section.
- For readers to quickly catch the contribution in this work, it would be better to highlight major difficulties and challenges, and authors' original achievements to overcome them, in a clearer way in Introduction section.
We modified these sentence in the introduction
Thus, there occur problem with accelerated wear resulting in downtime and loss of time for replacement for new one part. Any downtime can mean high economic costs and lower yields. The recipient of the tilling set is focused on its reliable and, preferably, maintenance-free use during the agrotechnical period.
And add these sentence in the introduction
The main challenge for fast-wearing elements is to find a treatment modifying the surface layer that would significantly extend the life of machine elements operating in very difficult conditions.
- It is suggested to highlight the limitations of this study, suggested improvements of this work and future directions in the conclusion section.
And add these sentence in the conclusions
The studies carried out also have some limitations. They were carried out in constant laboratory conditions, i.e. constant soil moisture and pH, constant speed and working depth of the tested samples, wall medium (in this case river sand) with a specific fraction. Field conditions vary considerably and depend on many factors, such as soil type, its compactness, moisture, contamination, soil ph, etc. Under such conditions, the wear intensity of the test samples could differ from the obtained laboratory results and would depend mainly on the previously mentioned soil parameters. Therefore, the directions of further research should take into account all these factors, and perhaps even be carried out in the field.
Another important factor that may limit the use of the aforementioned laser treatments is the cost of applying the appropriate Cr and Ni coating in laboratory conditions, which is several times higher than the purchase of a new spare part made of grey cast iron. If you want to introduce the indicated improvements to the coulters on an industrial scale and reduce the costs of modification of top layers, this process should be automated directly at the manufacturer of the spare part. However, this could present some difficulties due to the lack of appropriate technical facilities and the high cost of purchasing a laser coating apparatus.
- The properties of grey cast iron are well-established and no need to mention with a separate table. Therefore, Table 1 should be removed.
We removed the table 1
- Section 2 (Materials and methods) is described as point by point. It is not necessary to write the coating methods and their investigation procedure in point by point. It should be single paragraph (Descriptive).
- We have shortened this section (but also we needed to add some information because of comments of other Reviewers) and made single paragraph.
- Figure 2 is not mentioned in the manuscript. Similar mistakes have to be rectified throughout the manuscript.
We added a reference to figure 2 in the text
- The manuscript contains many SEM images, however, the detailed explanation about the inferences from the micrographs are missing. It should be discussed elaborately.
We have added more detailed explanation of SEM images.
A microscopy analysis of a cross section of the surface layer showed that after laser modification the alloyed zone appeared. It was observed in case of nickel implantation variant and as well as nickel and chromium. Such zone is characterize by very fine grained microstructure that it can be seen from the surface to the depth approx. 450 mm. in the figure 6 a and b. This is due to a lot of crystallisation nuclei that appear during rapid solidification from the liquid state. This process is happening during laser remelting without alloying. Such a microstructure is also much more homogeneous than microstructure of transformed ledeburite microstructure (visible in the figure 7) or core material microstructure consisted of flake graphite in the pearlite matrix.
More detailed SEM observation of alloyed zone microstructure in magnification of 2kx reviled characteristic for this kind of laser modification morphology which is based on the presence of a dendritic microstructure. in the whole modified zone. This is due to melting and solidification processes of the surface layer. Such process take place as well as in case of only remelting and also alloying. The difference between only remelting and alloying is possibility of forming some new phases containing implemented elements in case of alloying. Dendritic microstructure is visible in both cases of performed laser treatment variants (fig. 8a and 9a).
A microstructure study allowed to conclude, that during the second part of the modification solidification of surface layer, austenite dendrites are crystallizing directly from the liquid state of new mixed alloy (consisting of base material and alloying additions). As a consequence of rapid cooling hardening take place as usual in case of laser remelting only. Non-equilibrium solidification of new alloy takes place.
Higher magnification of modified layer (10kx) achieved by SEM analysis (fig. 8b and 9b) allowed to detect results of rapid solidification during laser treatment. Laser hardening caused formation of martensite phase. Martensite appears in case of laser remelting and in most cases of laser alloying – sometimes except implementing of cobalt for example [Paczkowska M., Makuch N., Kulka M: The influence of various cooling rates during laser alloying on nodular iron surface layer, Optics and Laser Technology - 2018, vol. 102, s. 60-67]. Martensite was especially clearly observed as a needles in dendritic microstructure (observed in figures with lower magnification – fig. 8a and 8b). Between martensite needles retained austenite is present.
- The visual findings of the SEM micrographs should be mentioned in the images itself. Unfortunately, none of the SEM images contains the inference. It should be incorporated.
We have added appropriate description in all SEM images.
- The results and discussion are not clearly dealt the outcomes of the proposed work. The authors should explicitly state the novel contribution of this work, the similarities and the differences of this work with the previous publications in this section.
We have improved the discussion about results and we have more emphasized the innovative nature of our research.
For example:
Similar microstructure was achieved after laser cladding using nickel on ductil cast iron [Rui Wang , Changyao Ouyang, Qihang Li, Qiaofeng Bai , Chunjiang Zhao and Yingliang Liu, Study of the Microstructure and Corrosion Properties of a Ni-Based Alloy Coating Deposited onto the Surface of Ductile Cast Iron Using High-Speed Laser Cladding, Materials 2022, 15, 1643.] that improved resistance to corrosion. But resistance to wear has not been studied.
or
For example after implementation of Cu-Ti-Ni by laser cladding on grey cast iron was improved 2-3-times in comparison to the material of the substrate [Lingjie Zhu, Yanhui Liu, Zhiwei Li, Lei Zhou, Yongjiu Li, Anhui Xiong, Microstructure and properties of Cu-Ti-Ni composite coatings on gray cast iron fabricated by laser cladding, Optics and Laser Technology 122 (2020) 105879].
or
In both cases roughness depth after laser treatment was smaller than roughness depth of untreated surface. The surface roughness is usually improved by laser alloying in case of treating such rough surface as cultivator coulters are characterized [Paczkowska M, Selech J. An Investigation of the Influence of Laser Alloying of the Surface Layer on Abrasive Wear Re-sistance of Cast Iron Elements. Tribologia 2018, 6, 107-117.]. Nevertheless, in more cases when the roughness of the base surface is small, laser remelting courses increase of roughness parameters - for example in research about laser nitriding od NiTi alloy [Hao Wang, Ralf Nett , Evgeny L. Gurevich, and Andreas Ostendorf , The Effect of Laser Nitriding on Surface Characteristics and Wear Resistance of NiTi Alloy with Low Power Fiber Laser, Appl. Sci. 2021, 11, 515. https://doi.org/10.3390/app11020515].
- Moreover, an extensive language editing is required to improve the quality of the manuscript. Authors are urged to rewrite the manuscript with a native English speaker.
The linguistic correctness has been corrected by the translator
Please note that the comments are intended merely to assist the authors in improving the manuscript and ensuring that published papers are of the highest quality. They are in NO WAY intended to discourage or demean the authors personally.

Reviewer 3 Report
This research of current, but some details need to be clarified in order to make the text clearer.
Material and methods:
- The text does not contain complete information about the casting sample. It is not clear whether the casting was subjected to any heat treatment? How were strength properties determined?
- It is not clear, the combination of nickel with chromium (in composition of 1:1) was stated in volume or weight fractions?
- What laser was used to treat the surface of the sample? What parameters of the laser track were set?
- The test device proposed in the work (Fig.2) does not provide the same conditions for exposure to an abrasive for all samples (Fig.3). For example, the sample closer to the rotation axis of the bowl will experience less impact than other samples due to the shorter way. This must be taken into account when interpreting test results.
Results and discussion
- When describing the results of the study, too many figures are used, some of them can be removed and replaced by other presentation. For example, figure 5a and figure 6 can be deleted. This is primary data, uninteresting to readers. Figure 13 can be replaced by a table.
- In Figures 9b, 10b, and 11, it would be good to indicate the observed phases on the microstructures.
- It would be nice to describe in more detail how the structure of cast iron changes under the influence of laser processing, not only with nickel or nickel and chromium additives, but also without additives. It is possible that the increase in wear resistance is associated with laser processing and can be achieved without the use of additives. If the manuscript presented comparative data on samples treated with a laser without the use of additives, this would be evidence of a positive influence of the use of nickel or nickel and chromium.
- The manuscript shows that nickel and chromium treatment does not change the weight loss after the wear test compared to nickel treatment only (Fig. 15). However, the surface layer hardness of samples modified with nickel and chromium is greater than that of samples modified only with nickel by about the same amount as the hardness of samples with nickel is greater than that of base samples (Fig. 14). Why is there such a difference in weight loss between the base and modified samples, but there is no difference between the samples modified with nickel and nickel and chromium? This issue needs to be discussed in more detail.
- It would be good to study the microstructure of samples after tribological tests.
- It is not clear how the surface roughness of the samples influence to wear resistance? Why is so much attention paid to the Ra determination in the manuscript?
Conclusion
- Written: "It was achieved by implementing elements such as nickel and chromium into the surface layer of cultivator cultures made of gray iron". This is not entirely correct. The surface layer of the castings has the structure of white cast iron. Therefore, the castings are not entirely made of gray cast iron.
- There is no explanation of the mechanism for increasing abrasion resistance after surface modification.
- The effect of laser treatment on the microstructure and composition of the surface layer has not been shown, although it has been studied.
- It is not clear which treatment is better with nickel or with nickel and chrome.
- It is not clear what values of surface roughness, hardness, microstructure do the authors consider optimal for good wear resistance? What laser treatment parameters allow them to be achieved?
Author Response
Dear Reviewer,
We would like to thank the Reviewer for valuable comments, which in our opinion contributed to increasing the quality of the paper. All suggestions and comments have been carefully thought out, and the corresponding revisions have been made to the paper. All changes in the article are marked blue red color in revision mode.
In addition, we have supplemented the incorporated changes with specific answers given below.
Yours sincerely
Authors
This research of current, but some details need to be clarified in order to make the text clearer.
Material and methods:
- The text does not contain complete information about the casting sample. It is not clear whether the casting was subjected to any heat treatment? How were strength properties determined?
We have added data about a heat treatment of flaps and about strength properties.
The material used for coulters flaps production is grey cast iron with flake graphite and mainly perlite in the matrix (Fig. 2). There was no through-heat treatment of coulters except surface treatment consisting on fast enough cooling process of their surface layer to create transformed ledeburite microstructure. Strength properties of this material are determined by a producer according to Standards.
- It is not clear, the combination of nickel with chromium (in composition of 1:1) was stated in volume or weight fractions?
We have added this information - in weight
- What laser was used to treat the surface of the sample? What parameters of the laser track were set?
We have added more detailed information about the kind of laser and the track.
The heating using a dual diode TRUDISK 1000 laser device was done in such way to remelt the surface layer of a piece of the cultivator coulter tip. It was taken into account that all coulter flap work at a depth about 5 cm ± 2 cm. The laser beam power fluence was 54J/mm2 and its collimation was equal to 12 mm. Such value of this parameter caused dimeter of laser track of 1.16 mm. The distance from the centers of particular laser tracks was 1.1mm. Laser beam scanned the surface to create an area of modified surface layer of approximately 160mm2 near the edge of the coulter.
- The test device proposed in the work (Fig.2) does not provide the same conditions for exposure to an abrasive for all samples (Fig.3). For example, the sample closer to the rotation axis of the bowl will experience less impact than other samples due to the shorter way. This must be taken into account when interpreting test results.
We added the explanation in the article
Because of there was a difference in linear velocities at various distances from the rotation axis of the bowl, the arm holding the test samples was inverted half way through the test, and the rotation direction was changed as well. This resulted in the wear of each of the three samples tested simultaneously was more uniform. In order to reflect actual working conditions as closely as possible, the samples were placed at intervals of approx. 12 cm from one another to ensure that the abrasive medium flowing over the sample is not affected by the adjacent sample.
Results and discussion
- When describing the results of the study, too many figures are used, some of them can be removed and replaced by other presentation. For example, figure 5a and figure 6 can be deleted. This is primary data, uninteresting to readers. Figure 13 can be replaced by a table.
We deleted the figures 5a ana 6 and replaced figure 13 by a table
- In Figures 9b, 10b, and 11, it would be good to indicate the observed phases on the microstructures.
We have done it (the numbers of figure was changed).
- It would be nice to describe in more detail how the structure of cast iron changes under the influence of laser processing, not only with nickel or nickel and chromium additives, but also without additives. It is possible that the increase in wear resistance is associated with laser processing and can be achieved without the use of additives. If the manuscript presented comparative data on samples treated with a laser without the use of additives, this would be evidence of a positive influence of the use of nickel or nickel and chromium.
We have done the more detailed description about forming the microstructure.
A microscopy analysis of a cross section of the surface layer showed that after laser modification the alloyed zone appeared. It was observed in case of nickel implantation variant and as well as nickel and chromium. Such zone is characterize by very fine grained microstructure that it can be seen from the surface to the depth approx. 450 mm. in the figure 6 a and b. This is due to a lot of crystallisation nuclei that appear during rapid solidification from the liquid state. This process is happening during laser remelting without alloying. Such a microstructure is also much more homogeneous than microstructure of transformed ledeburite microstructure (visible in the figure 7) or core material microstructure consisted of flake graphite in the pearlite matrix.
More detailed SEM observation of alloyed zone microstructure in magnification of 2kx reviled characteristic for this kind of laser modification morphology which is based on the presence of a dendritic microstructure in the whole modified zone. This is due to melting and solidification processes of the surface layer. Such process take place as well as in case of only remelting and also alloying. The difference between only remelting and alloying is possibility of forming some new phases containing implemented elements in case of alloying. Dendritic microstructure is visible in both cases of performed laser treatment variants (Fig. 8a and 9a).
Higher magnification of modified layer (10kx) achieved by SEM analysis (fig. 8b and 9b) allowed to detect results of rapid solidification during laser treatment. Laser hardening caused formation of martensite phase. Martensite appears in case of laser remelting and in most cases of laser alloying – sometimes except implementing of cobalt for example [Paczkowska M., Makuch N., Kulka M: The influence of various cooling rates during laser alloying on nodular iron surface layer, Optics and Laser Technology - 2018, vol. 102, s. 60-67]. Martensite was especially clearly observed as a needles in dendritic microstructure (observed in figures with lower magnification – fig. 8a and 8b). Between martensite needles retained austenite is present.
Referring to increasing wear resistance by only laser remelting – yes, you are absolutely right. It is possible to increase it by only remelting the surface layer. We have done it. Wear results could be comparable to the results after laser alloying in some cases. But by applying some alloying elements also some other properties could be improved like corrosion resistance which is also important taking into account of work conditions of cultivator coulters. The first step of this research was to check if laser alloying can increase the wear resistance (in some case – as we showed in the introduction – alloying decreasing wear resistance – the example of chromium [da Costa A.R., Vilar R. Erosion by solid particle impingement: experimental results with cast-iron, laser-treated surface, Tribology Letters 3(4), 1997, 379–385]). The next step we are going to do is check the corrosion resistance.
- The manuscript shows that nickel and chromium treatment does not change the weight loss after the wear test compared to nickel treatment only (Fig. 15). However, the surface layer hardness of samples modified with nickel and chromium is greater than that of samples modified only with nickel by about the same amount as the hardness of samples with nickel is greater than that of base samples (Fig. 14). Why is there such a difference in weight loss between the base and modified samples, but there is no difference between the samples modified with nickel and nickel and chromium? This issue needs to be discussed in more detail.
Wear resistance is not only function of hardness. A great importance is microstructure morphology. Microstructure after laser remelting and alloying is much more fine grained with less sharp ends of grains that are stresses concentrators. We expanded the clarification.
The mass loss reduction of the treated parts working in fruition conditions is a consequence not only of increased surface hardness but also a result of a significantly modified, more uniform microstructure with small grains with less sharp ends of grains that are stresses concentrators. Hence, quite high hardness of surface layer of cultivator coulters made of transformed ledeburite (not laser treated) does not turn into proportionally greater wear resistance
- It would be good to study the microstructure of samples after tribological tests.
Yes, it is a right comment. We are going to perform such studies in the next step of our research but in this stage we based on mass result and the surface condition. We expect that in the surface layer microstructure we will find deformed grains and some particles of sand.
- It is not clear how the surface roughness of the samples influence to wear resistance? Why is so much attention paid to the Ra determination in the manuscript?
Ra parameter is the most common surface roughness parameter, representing the arithmetic roughness average of the surface. The majority of the methods for linking surface fatigue or failure to surface roughness use just Ra.
Conclusion
- Written: "It was achieved by implementing elements such as nickel and chromium into the surface layer of cultivator cultures made of gray iron". This is not entirely correct. The surface layer of the castings has the structure of white cast iron. Therefore, the castings are not entirely made of gray cast iron.
You are right. We have corrected it.
It was achieved by implementing elements such as nickel and chromium into the surface layer that was originally made of white cast iron.
- There is no explanation of the mechanism for increasing abrasion resistance after surface modification.
We added the explanation in the text
The surface laser treatment increased the surface roughness grey cast iron due to the laser ablation by the instant high temperature of a laser beam. The microhardness and elemental composition along the cross-section in both cases (Ni and Ni+ Cr) were also analysed, which shows that the hardness was increased after laser modification treatment. The friction coefficient and wear resistance of grey cast iron were also improved after laser modification.
- The effect of laser treatment on the microstructure and composition of the surface layer has not been shown, although it has been studied.
We have emphasized those aspects in the conclusions
….- a modified area after laser alloying was showed after nickel as well as nickel and chromium implementation was characterized by hardened microstructure (with martensite), higher homogeneity and fine grains,
- nickel as well as nickel and chromium presence in laser modified surface layer has been proved,…
- It is not clear which treatment is better with nickel or with nickel and chrome.
On the base of ours result there was no difference between those two applied variants. So there is no need to add chromium. We have used chromium to one variant because we wanted to check the influence of chromium addition, because based on the literature there is some differences in results after alloying of chromium using laser treatment. Although chromium increases hardness, corrosion resistance not always increases the wear resistance, as was noticed in [da Costa A.R., Vilar R. Erosion by solid particle impingement: experimental results with cast-iron, laser-treated surface, Tribology Letters 3(4), 1997, 379–385].
We add sentence in “Conclusions” to underline this effect.
As results showed there is no difference in wear resistance between implementation of only nickel and nickel and chromium. Consequently it could be stated that there is no need to add chromium in to the surface layer of culture (except possibility of increasing corrosion resistance if it would be proved in such case) .
- It is not clear what values of surface roughness, hardness, microstructure do the authors consider optimal for good wear resistance? What laser treatment parameters allow them to be achieved?
It seems that hardness above 800 HV0.1 is enough to achieve quite good wear resistance results. More important is to create uniform and fine crystalline microstructure, as was achieved in presented research.
There is no need to achieve smooth surface in case of parts working in soil. Therefore, super finishing process in case of cultivator coulter is not required. As was appeared, laser remelting in case of analysing machine parts even decreases surface roughness.
Performed laser treatment parameters such as 54J/mm2 of the laser beam power fluence and 12 mm of its collimation was suitable to achieve modified surface layer that increases wear resistance.
So, we have added information in the conclusion about those values:
It could be conclude that hardness above 800 HV0.1 seems to be enough to achieve quite good wear resistance results. More important is to create uniform and fine crystalline microstructure, as was achieved in presented research. As appeared laser alloying in case of analysing machine parts decreases surface roughness because of high roughness of base surface. But it is worthy underline that there is no need to achieve smooth surface in case of parts working in soil. Therefore, super finishing process in case of cultivator coulter is not required.
Performed laser treatment parameters such as 54J/mm2 of the laser beam power fluence and 12 mm of its collimation was suitable to achieve modified surface layer that increases wear resistance.

Reviewer 4 Report
Review of manuscript
The analysis of the impact of laser surface modification with Ni and Cr implementation on the microstructure of grey iron and its tribological behavior
The paper topic is actual, and the results are promising. However, the description and analysis of experimental data have a lot of mistakes and are not clear. The structure examination and analysis need to be considerably improved (see some comments below). English is not proper. So, unfortunately, I cannot recommend manuscript for publication. Intensive rewriting is needed
Comments:
- Lines 22-23: The sentence “The wear test (performed on a specially designed device for testing tribology properties in different mediums) showed that such modification increases resistance to wear” is not completely clear. The question is : What modification parameters allows to increase the wear resistance?
- Lines 35-36: Sentence ”… there occur problem with accelerated wear resulting in costs related with buying new elements and with downtimes required for exchanging of worn parts” is not clear . Please, edit
- Lines 45-57: The description “… This treatment could be performed with such parameters of laser beam to remelt some part of the layer and implement simultaneously some alloying elements or even compounds into it. Microstructure of layer, formed as a result of such modification, is usually very fine, contains martensite, and sometimes also some new phases could be appeared (as a consequence of reaction between base material and implemented substances). Some similarities to ledeburite type of microstructure could be also be found, like primary cementite needles. There are many examples of effective surface modification of grey irons using laser beam. For instance, laser implementation of carbon, boron, tungsten and chromium to the surface layer of nodular iron caused formation a microstructure rich in carbides with 1200 HV0.05 of hardness [16]. Hardness as well as resistance to corrosion was found after copper implantation into the cast iron according to Zeng D. et. al [17]. Such properties as fatigue resistance could also be increase using laser modification” is not clear and contains the English mistakes (underlined)
- Lines91-93: The statement “…The microstructure consists of flake graphite and mainly perlite in the matrix but surface layer of ready coulter contains transformed ledeburite microstructure as a result of chilling” is not correct. You need to define parameter “chilling speed” which influences the structure formation. Please, correct. Additionally, microstructure of base material needs to be shown.
- Lines 95-96: English of the sentence “….The first step of the modification treatment consisted in applying an appropriate covering coating (the coating alloying is contacting containing two ingredients: element or elements and bounding substance) on the surface of the piece” is not proper. What does it mean “ covering coating (the coating alloying is contacting containing two ingredients: element or elements and bounding substance)” ?
- Line 101: There is no description of coatings preparation
- Lines 105-111: There is no detailed description of laser processing. Authors only said: “ A 110 TRUDISK 1000 laser device was used. The laser beam power fluence was 54J/mm2 and its 111 collimation was equal to 12 mm.” It is not enough
- Lines 107_108: How to understand (underlined): “….It was taken into account that all parts of machines working in the ground including the coulter flap work at a depth from 2 to 5 cm”? It is impossible.
- Line158: What does it mean”… stereometric structure of the surface quality”?
- Lines 157-165: The explanation is not clear. For example, it is seen from the sentence “…The mean roughness depth (Rz) for untreated surface was 55μm, after nickel implementation it was 17,5μm and after implementation of a combination of nickel and chromium it was also smaller - 21μm.” Rz =21 μm is less than Rz =17.5 μm. How to understand? Intensive rewriting is needed
- Lines 176-178: Authors state “…. A microscopy analysis of a cross section of the surface layer showed that after laser modification a much more uniform microstructure surface layer (in contrast to the core mate- 177 rial) could be observed (Fig. 7).” It is impossible to evaluate the uniformity of microstructure at images with low magnification-100x (Fig.7). What is the parameter “uniformity of microstructure”?
- Lines 204-205: Authors statement “…It is worth emphasising that in comparison to the surface layer only after chilling where a white cast iron microstructure (Fig. 11) is appearing, layer modified by laser beam is much more uniform and grains are much more smaller.” is not clear and correct. It is well known that cast iron microstructure depends on cooling rate, and authors need to define influence of cooling rate on the austenite transformation.
- Line 217: It is not X-ray analysis. It is EDS analysis (Fig.12).
- Lines 230-234: Authors need to explain why Ni and Cr additions increase the hardness of surface layer. What phases are formed? Microhardness of each phases needs to be examined
- Lines 235-255: Authors did not present any possible wear mechanisms resulting in increase of wear resistance
- Line272: Conclusion needs to be corrected taking into account the comments above
Author Response
Dear Reviewer,
We would like to thank the Reviewer for valuable comments, which in our opinion contributed to increasing the quality of the paper. All suggestions and comments have been carefully thought out, and the corresponding revisions have been made to the paper. All changes in the article are marked by red color.
In addition, we have supplemented the incorporated changes with specific answers given below.
Yours sincerely
Authors
Review of manuscript
The analysis of the impact of laser surface modification with Ni and Cr implementation on the microstructure of grey iron and its tribological behaviour
The paper topic is actual, and the results are promising. However, the description and analysis of experimental data have a lot of mistakes and are not clear. The structure examination and analysis need to be considerably improved (see some comments below). English is not proper. So, unfortunately, I cannot recommend manuscript for publication. Intensive rewriting is needed
Comments:
- Lines 22-23: The sentence “The wear test (performed on a specially designed device for testing tribology properties in different mediums) showed that such modification increases resistance to wear” is not completely clear. The question is : What modification parameters allows to increase the wear resistance?
We change the sentence and add an explanation:
The wear test showed that Ni and Cr laser coatings increased resistance to abrasive wear resulting from the modification of the microstructure by the formation of martensite and grain frag-mentation.
- Lines 35-36: Sentence ”… there occur problem with accelerated wear resulting in costs related with buying new elements and with downtimes required for exchanging of worn parts” is not clear . Please, edit
We change the sentence.
Thus, there occur problem with accelerated wear resulting in downtime and loos of time for replacement for new one part. Any downtime can mean high economic costs and lower yields. The recipient of the tilling set is focused on its reliable and, preferably, maintenance-free use during the agrotechnical period. The main challenge for fast-wearing elements is to find a treatment modifying the surface layer that would significantly extend the life of machine elements operating in very difficult conditions.
- Lines 45-57: The description “… This treatment could be performed with such parameters of laser beam to remelt some part of the layer and implement simultaneously some alloying elements or even compounds into it. Microstructure of layer, formed as a result of such modification, is usually very fine, contains martensite, and sometimes also some new phases could be appeared (as a consequence of reaction between base material and implemented substances). Some similarities to ledeburite type of microstructure could be also be found, like primary cementite needles. There are many examples of effective surface modification of grey irons using laser beam. For instance, laser implementation of carbon, boron, tungsten and chromium to the surface layer of nodular iron caused formation a microstructure rich in carbides with 1200 HV0.05 of hardness [16]. Hardness as well as resistance to corrosion was found after copper implantation into the cast iron according to Zeng D. et. al [17]. Such properties as fatigue resistance could also be increase using laser modification” is not clear and contains the English mistakes (underlined)
We corrected these sentences:
Laser processing allows the surface layer to be melted down with the simultaneous addition of alloying elements to it. The resulting microstructure is usually highly fragmented and often contains martensite. In some cases it is similar to ledeburite, for example it contains primitive cementite spines. In the literature, there are many examples of grey cast iron surface modification with the use of a laser beam, which significantly changes the selected parameters of the surface layer. The laser implementation of selected elements, i.e. carbon, boron, tungsten or chromium on the surface of nodular iron, resulted in the formation of a microstructure containing a lot of carbides with a very high hardness reaching 1200 HV0.05 [16]. The increase in hardness and increased corrosion resistance were also found after copper implantation in cast iron [17].
- Lines 91-93: The statement “…The microstructure consists of flake graphite and mainly perlite in the matrix but surface layer of ready coulter contains transformed ledeburite microstructure as a result of chilling” is not correct. You need to define parameter “chilling speed” which influences the structure formation. Please, correct. Additionally, microstructure of base material needs to be shown.
By “chilling” we meant fast enough cooling to avoid graphite forming in the surface layer of microstructure of grey iron. Consequently, in the text we have corrected on:
“…surface treatment consisting on fast enough cooling process of their (coulters) surface layer to create transformed ledeburite microstructure.”
We have added the microstructure.
- Lines 95-96: English of the sentence “….The first step of the modification treatment consisted in applying an appropriate covering coating (the coating alloying is contacting containing two ingredients: element or elements and bounding substance) on the surface of the piece” is not proper. What does it mean “ covering coating (the coating alloying is contacting containing two ingredients: element or elements and bounding substance)” ?
Yes, “contacting containing” is a mistake. We have corrected it.
Covering coating is a mixture consisting of alloying element/elements, water glass and distillated water.
- Line 101: There is no description of coatings preparation
We supplemented the description. We have added information about composition of chromium and nickel that there is in weight and added that “Coating was mixed of element (or elements), water glass as a bounding substance and distillated water.”
- Lines 105-111: There is no detailed description of laser processing. Authors only said: “ A 110 TRUDISK 1000 laser device was used. The laser beam power fluence was 54J/mm2 and its 111 collimation was equal to 12 mm.” It is not enough
We added more information
The heating using a dual diode TRUDISK 1000 laser device was done in such way to remelt the surface layer of a piece of the cultivator coulter tip. It was taken into account that all coulter flap work at a depth about 5 cm ± 2 cm. The laser beam power fluence was 54J/mm2 and its collimation was equal to 12 mm. Such value of this parameter caused dimeter of laser track of 1.16 mm. The distance from the centers of particular laser tracks was 1.1mm. Laser beam scanned the surface to create an area of modified surface layer of approximately 160mm2 near the edge of the coulter.
- Lines 107_108: How to understand (underlined): “….It was taken into account that all parts of machines working in the ground including the coulter flap work at a depth from 2 to 5 cm”? It is impossible.
We changed the sentence
It was taken into account that all coulter flap work at a depth about 5 cm ± 2 cm.
- Line158: What does it mean”… stereometric structure of the surface quality”?
We change the sentence
Examples of the surface profiles of a modified part are presented in the Fig. 4. As could be noticed, such treatment can change the stereometric structure of the surface quality of the cultivator coulter.
- Lines 157-165: The explanation is not clear. For example, it is seen from the sentence “…The mean roughness depth (Rz) for untreated surface was 55μm, after nickel implementation it was 17,5μm and after implementation of a combination of nickel and chromium it was also smaller - 21μm.” Rz =21 μm is less than Rz =17.5 μm. How to understand? Intensive rewriting is needed
Yes you are right. It clear. We meant that Rz (21μm) after implementation of a combination of nickel and chromium is also smaller than Rz for untreated surface (55μm). We have rewrote that.
The mean roughness depth (Rz) for untreated surface was 55μm, after nickel implementation it was 17,5μm and after implementation of a combination of nickel and chromium was 21μm. In both cases roughness depth after laser treatment was smaller than roughness depth of untreated surface.
- Lines 176-178: Authors state “…. A microscopy analysis of a cross section of the surface layer showed that after laser modification a much more uniform microstructure surface layer (in contrast to the core material) could be observed (Fig. 7).” It is impossible to evaluate the uniformity of microstructure at images with low magnification-100x (Fig.7). What is the parameter “uniformity of microstructure”?
Yes, it could be not clear. We have marked the laser modified surface layer and the transformed ledeburite zone in this figure (now is no 6). We think, that know (even in this case of magnification) the difference between those areas should be more evident. More clear difference in “uniformity of microstructure” is between figures 8a, 9a where we have showed microstructure after laser modification and figure 10 where we have showed the transformed ledeburite zone in higher magnification (both in 2000x).
We did not assessed uniformity of microstructure quantitatively. So we did not use parameters. We only have showed that in comparison of transformed ledeburite zone microstructure, the homogeneity of laser modified layer microstructure is higher.
- Lines 204-205: Authors statement “…It is worth emphasising that in comparison to the surface layer only after chilling where a white cast iron microstructure (Fig. 11) is appearing, layer modified by laser beam is much more uniform and grains are much more smaller.” is not clear and correct. It is well known that cast iron microstructure depends on cooling rate, and authors need to define influence of cooling rate on the austenite transformation.
We rewrote this statement.
SEM observation of surface layer of no laser treated coulter in higher magnification (2kx) that can be seen in the figure 10 reviled transformed ledeburite microstructure characterized for white cast iron. Such microstructure consists of pearlite (visible as a cementite and ferrite plates) and eutectic. It is worth emphasising that in comparison to the surface layer after laser treatment this microstructure is much more coarse-grained and heterogeneous. Additionally grains are characterized by sharp ends which is not conducive to wear resistance.
You are right, generally not only in case of cast iron microstructure depends on cooling rate. However, the aim of chilling process (that is commonly applied in case of surface layer of grey irons parts) is to form hard transformed ledeburite microstructure (no martensite). The microstructure that is achieved is pearlite and eutectic (consisted of pearlite and cementite after eutectoid transformation). So, cooling rate during chilling is low enough to form rather stable phase not metastable as martensite.
- Line 217: It is not X-ray analysis. It is EDS analysis (Fig.12).
We corrected it
- Lines 230-234: Authors need to explain why Ni and Cr additions increase the hardness of surface layer. What phases are formed? Microhardness of each phases needs to be examined
- Lines 235-255: Authors did not present any possible wear mechanisms resulting in increase of
wear resistance
We added description of wear mechanism
Abrasive wear was the main mechanism for material wear in soil. Soil was mainly containing sand fraction from 0,2 to 0,3 mm about 90 % and smaller fraction from 0,01 to 0,2 about 8% and rest was dusty fraction. An increased portion of coarse grains was responsible for microcutting and grooving, and portion of fine fractions for microcutting.
- Line272: Conclusion needs to be corrected taking into account the comments above
We have corrected the conclusion taking into account all comments from reviewers

Round 2
Reviewer 1 Report
The Author responded positively to changing the points
Reviewer 2 Report
The manuscript is now can be accepted for the publication.
Reviewer 3 Report
I think the manuscript can be accepted as is. However, you can improve fig. 2 if you make the ruler and the inscription white color.
Reviewer 4 Report
no comments